# Towards Foveated Rendering For Immersive Remote Telerobotics

Y. T. Tefera[1,2] (iD), D. Mazzanti[1] (iD), S. Anastasi[3] (iD), D. G. Caldwell[1] (iD), P. Fiorini[2] (iD), and N. Deshpande[1] (iD)

[1]Istituto Italiano di Tecnologia (IIT), Via Morego 30, 16163 Genova, Italy
[2]Department of Computer Science, University of Verona, Via Strada Le Grazie 15, 37134 Verona, Italy
[3]Istituto Nazionale per l'Assicurazione contro gli Infortuni sul Lavoro (INAIL), P.le Pastore 6, 00144 Rome, Italy

*Abstract*—In remote telerobotics applications, real-time 3D perception through RGB-D cameras (real-time 3D reconstruction, point-clouds), and its rendering inside modern virtual reality (VR) environments, can enhance a user's sense of presence and immersion in a remote scene. However, this approach requires that the whole pipeline from sensor data acquisition to VR rendering satisfy the speed, throughput, and visual quality requirements. Point-cloud data suffers from network latency and throughput limitations that can negatively impact user experience. In this research, the human visual system was taken as an inspiration to address this problem. Human eyes have their sharpest visual acuity at the center of their field-of-view, which falls off at the periphery. A remote 3D data visualization framework is proposed that utilizes this acuity fall-off to facilitate the processing, transmission, buffering, and rendering in VR of dense point-clouds / 3D reconstructed scenes. The proposed framework shows significant reductions in latency and throughput needs, higher than 60% in both. A preliminary user study shows that the framework does not significantly affect the perceived visual quality.

*Index Terms*—3D Reconstruction, Virtual Reality, Gaze tracking, Foveated Rendering, Telerobotics

## I. INTRODUCTION

Remote telerobotics applications have received increased interest in recent times due in no small measure to the ongoing COVID-19 pandemic. Effective implementations in this field would immeasurably improve the lives of frontline workers, being able to respond to certain emergencies without requiring physical presence [21]. The advances in the field are especially attributed to the ready availability of good quality, low-cost, consumer-grade sensors (RGB-D cameras), and immersive virtual reality (VR) devices [22]. This has helped the development of novel algorithms in real-time point-cloud acquisition and 3D scene reconstruction [7, 19]. *Immersive remote telerobotics* (IRT), i.e., the combination of VR and real-time 3D visual data from remote RGB-D cameras can allow real-time immersive visualization and interaction by the user, perceiving the colour and 3D profile of the remote scene and robotic agents simultaneously [17, 10]. The user can experience enhanced situational awareness while maintaining their presence illusion [16, 20]. This combination is the key distinguishing factor against traditional teleoperation interfaces, which rely on mono- or stereo-video feedback and suffer from limitations in terms of fixed or non-adaptable camera viewpoints, occluded views of

This research was conducted in collaboration with the Italian National Worker's Compensation Authority (INAIL).

the remote space, etc. [2, 8]. Nevertheless, the increased data footprint (3D vs 2D) in real-time IRT imposes constraints on resolution, latency, throughput, compression, acquisition, and the visualization of this information [16, 12]. For instance, latency and low resolution negatively impact the sense of presence and provoke cybersickness [9, 15].

In this paper, the human visual system serves as the inspiration to address the coupling between the 3D data acquisition and its rendering in IRT. The human eye has the highest visual acuity at the center of its field-of-view, and this acuity falls off towards the periphery [5]. This acuity fall-off can facilitate the processing, streaming, and rendering of 3D data to a remote user in VR, thereby optimizing the amount of data transmitted. The user's gaze is exploited to divide the acquired 3D data into concentric conical regions of progressively reducing resolution away from the center of the gaze. It is shown that such data manipulation offers significant benefits in latency and throughput. Preliminary analysis shows that it has minimal impact on the perceived visual experience for the user.

## II. SYSTEM OVERVIEW

### A. Human Visual Acuity and Foveation

Humans perceive visual information through two kinds of photoreceptors in the retina: cones and rods. As shown in Figure 1-A the cone density is highest in the central region of the retina, and reduces monotonically to a reasonably even density into the peripheral retina region. This distribution is the concept of *Foveation*. Retinal eccentricity is the angle at which light from a scene / image gets focused on the retina. With the photoreceptors' density reducing monotonically, it is possible to approximate the retina as being formed of discrete concentric regions. The density of the photoreceptors is inversely proportional to the eccentricity angles [11]. Table I gives an example of such an approximation for retinal regions. Figure 1-B shows an example of how the concentric regions are applied to *foveate* the point-cloud.

*1) Visual Acuity:* is quantitatively represented in terms of the *minimum angle of resolution* (MAR, measured in arcminutes), which is the smallest angle at which two objects in the visual scene are perceived as separate by the human eye [18]. The relationship between MAR and eccentricity can be approximated as a linear model, Eq. 1. This has been shown to closely match the anatomical features of the eye [18, 3].

TABLE I
HUMAN RETINAL REGIONS AND THEIR SIZES IN DIAMETER AND
ECCENTRICITY ANGLE (DERIVED FROM [11]).

| | Region | Diameter (mm) | Eccentricity $^\circ$ |
|---|---|---|---|
| $R_0$ | Fovea | 1.5 | $5^\circ$ |
| $R_1$ | ParaFovea | 2.5 | $8^\circ$ |
| $R_2$ | PeriFovea | 5.5 | $18^\circ$ |
| $R_3$ | Near Peripheral | 8.5 | $30^\circ$ |
| $R_4$ | Mid Peripheral | 14.5 | $60^\circ$ |
| $R_5$ | Far Peripheral | 26 | $> 60^\circ$ |

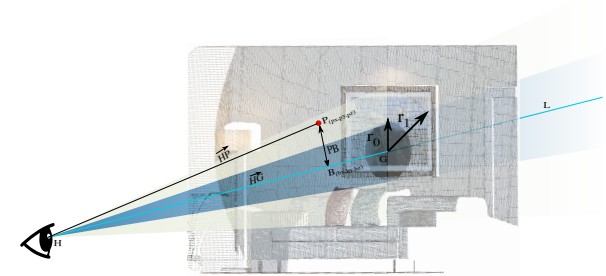

Fig. 2. Map partitioning - the surfel point $\mathbf{P}(px, py, pz)$ is classified using the ray $\mathbf{L}$ cast from the point of gaze origin $\mathbf{H}(hx, hy, hz)$.

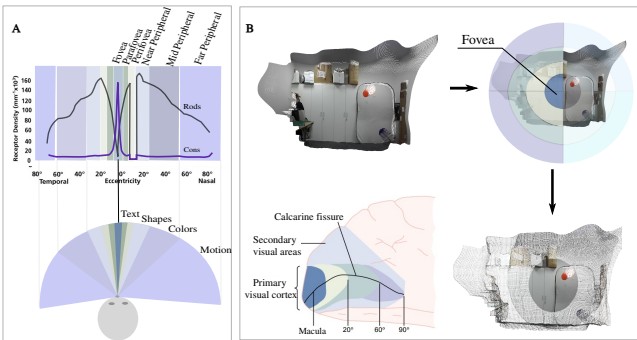

Fig. 1. A) Rods and Cones distribution in the retina. B) Retinotopic organization of the primary visual cortex (bottom-left). Foveation applied to a sample point-cloud (bottom-right).

used as a point of origin $\mathbf{H}(hx, hy, hz)$. A ray is cast from $\mathbf{H}(hx, hy, hz)$, i.e., the *gaze vector* $\mathbf{L} \in \mathbb{R}^3$, and is extended up to the last point of intersection $\mathbf{G}(gx, gy, gz)$ with the surfel map $\mathcal{M}$. The foveation regions are now structured around $\mathbf{L}$. To assign each surfel in $\mathcal{M}$ to a particular region $\mathcal{M}_n$ ($n \in \{0...N\}$ retinal regions), the shortest distance, i.e., the perpendicular distance between the surfel $\mathbf{P}(px, py, pz)$ and $\mathbf{L}$, $\mathbf{PB} \perp \mathbf{L}$, where $\mathbf{B}(bx, by, bz)$ is a point on $\mathbf{L}$ is used, as shown in Fig. 2. The algorithm 1 is implemented in CUDA in the GPU for faster processing.

$$MAR = mE + MAR_0 \tag{1}$$

Here $MAR_0$ is the intercept, which signifies the smallest resolvable eccentricity angle for humans, and $m$ is the slope of the linear model. Authors in [3] experimentally determined the values of $m$ based on observed image quality, ranging between 0.022 to 0.034. This formulation applies the concept of foveation to RGB-D data.

### B. Real-time 3D Data Acquisition and Foveated Sampling

The acquired RGB and depth map images from the RGB-D camera are utilized in two ways: (i) as a point-cloud represented as an unordered list of *surfels*, where each surfel has a position $\mathbf{p} \in \mathbb{R}^3$, a normal $\mathbf{n} \in \mathbb{R}^3$, a colour $\mathbf{c} \in \mathbb{R}^3$, a weight $w \in \mathbb{R}$, a radius $r \in \mathbb{R}$, an initialization timestamp $t_0$, and a current timestamp $t$; and (ii) the mapping pipeline uses the state-of-the-art dense visual SLAM system, ElasticFusion [19], at each time step $t$, to register the colour image $\mathbf{C}_t$ and the depth map $\mathbf{D}_t$ into the global 3D reconstruction map, $\mathcal{M}$, by estimating the camera pose. The alignment is achieved by minimizing the geometric and photometric error [19].

*1) Map Partitioning:* For brevity, the symbol $\mathcal{M}$ is used interchangeably for the real-time point-cloud and the global 3D reconstruction map. Applying the foveation model to $\mathcal{M}$ implies projecting the retinal fovea regions into it to partition it into concentric conical regions. $\mathcal{M}$ is then resampled to approximate the monotonically decreasing visual acuity in the foveation model, termed *foveated sampling*.

To partition $\mathcal{M}$ into regions, the eye gaze direction and its point of origin are utilized. The center of the eye gaze is

---

**Algorithm 1:** Map Partitioning Algorithm

```
Input:  M          /* Map to be partitioned  */
        L          /* Gaze direction vector   */
        e_0 ...n       /* Eccentricity angles     */
foreach  surfel P_i in the map M do
    B ← proj_L^{P_i}      /* projection of P_i on L */
    d^{vi} ← ||HB||    /* dist. between H and B */
    d ← PB⊥L         /* shortest distance */
    for j=1 to max(e) do
        r_j ← tan(e_j) * d^{vi}   /* calc. radii r_j   */
    end
    /* put P_i into the maps M_0 ...M_n     */
    if d < r_0 then
        M_0 ← P_i;
    else if d > r_0 AND d ≤ r_1 then
        M_1 ← P_i;
    ⋮
    else
        M_n ← P_i
    end
end
```

*2) Foveated PCL Sampling:* The partitioned global map $\mathcal{M}$, with the region-assigned surfels, then needs to be downsampled to follow the acuity drop-off, as seen in Fig. 1. For this, $\mathcal{M}$ is converted into a PCL point-cloud data structure, $\mathcal{P}_n$ for each $\mathcal{M}_n$ region $\forall n \in \{0...N\}$. To implement the foveated sampling, the $\mathbb{R}^3$ space of each $\mathcal{P}_n$ region needs to be further partitioned into an axis-aligned regular grid of cubes as shown in Fig. 3. This process of re-partitioning the regions is called *voxelization* [13] and the discrete grid elements are called *voxels*.

This voxelization and down-sampling is a three-step pro-

cess: (1) calculating the volume of the voxel grid in each region, which is the point-cloud distribution along x-, y-, and z-axes; (2) calculating the voxel size, i.e., dimension, $\mathfrak{v}_n$, for the voxelization in each region, and (3) down-sampling by approximating the point-cloud inside each voxel by its 3D centroid point.

For the voxel size, $\mathfrak{v}$, consider the voxelization of the central fovea region, $\mathcal{P}_0$. The smallest angle a healthy human with a normal visual acuity of 20/20 can discern is 1 arcminute, i.e., $0.016667°$. In Eq. (1) therefore, $MAR_0 = 0.016667°$. Eq. 2 calculates the smallest visually resolvable object length.

$$\mathfrak{l} = d^{vi} * \tan(MAR_0) \tag{2}$$

The important consideration here is the value of $d^{vi}$, which is the distance to the image along the gaze vector $\mathbf{L}$. In Alg. 1, a $d^{vi}$ value for each surfel is calculated. In contrast, here in order to down-sample the region based on the voxelization, we calculate one $d^{vi}$ value for the entire $\mathcal{P}_0$ region, approximated as the distance from $\mathbf{H}(hx, hy, hz)$ to the 3D centroid of the point-cloud in the region, Eq. (3).

$$\mathfrak{pc}_0 = \frac{1}{N_{\mathcal{P}_0}}\left(\sum_{i=1}^{N_{\mathcal{P}_0}} x_i, \sum_{i=1}^{N_{\mathcal{P}_0}} y_i, \sum_{i=1}^{N_{\mathcal{P}_0}} z_i\right) \tag{3}$$

$$d_0^{vi} = \mathbf{d}(\mathbf{H}, \mathfrak{pc}_0) \tag{4}$$

, where $N_{\mathcal{P}_0}$ is the number of PCL points in $\mathcal{P}_0$, and $\mathbf{H}$ is the eye gaze origin. Then, Eq. (2) is re-written as Eq. (5) to give the voxel size $\mathfrak{v}_0$ for the region.

$$\mathfrak{v}_0 = d_0^{vi} * \tan(MAR_0) \tag{5}$$

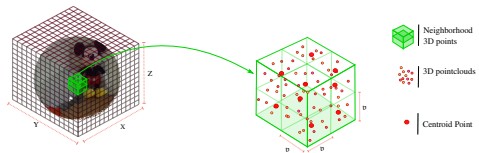

Fig. 3. PCL voxelization - the point-cloud inside each voxel is approximated by its centroid in that voxel.

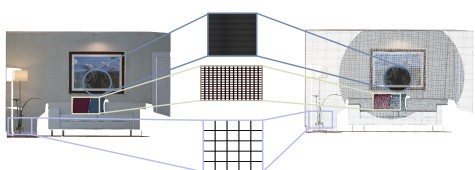

Fig. 4. Foveated point-cloud sampling example showing the different voxel grid sizes for the different regions.

Once the voxelization of region $\mathcal{P}_0$ is finalized, for the subsequent concentric regions from $\mathcal{P}_1$ to $\mathcal{P}_n$, the voxel sizes are correlated through the linear MAR relationship. Eq. (6) shows that as the eccentricity angle of the regions increases, so do the voxel sizes.

$$MAR_n = m \cdot E_n + MAR_0$$
$$\mathfrak{v}_n = \frac{MAR_n}{MAR_{n-1}} * \mathfrak{v}_{n-1} \tag{6}$$

The increasing voxel size away from the fovea region implies more and more surfels of the point-cloud of the corresponding regions are now accommodated within each single voxel of that region. Therefore, when the down-sampling step is applied, the approximation of the point-cloud within a voxel is done over progressively dense voxels. For the down-sampling part, the region $\mathcal{P}_0$ being the fovea region is left untouched so its density is the same as the incoming global map density, i.e., the resolution set for the RGB-D camera. The down-sampling in the subsequent regions is done by approximating the point-cloud within each voxel with its 3D centroid, using Eq. (7).

$$\mathfrak{pc}_n^v(x, y, z) = \frac{1}{N_{\mathcal{P}_n}^v}\left(\sum_{i=1}^{N_{\mathcal{P}_n}^v} x_i, \sum_{i=1}^{N_{\mathcal{P}_n}^v} y_i, \sum_{i=1}^{N_{\mathcal{P}_n}^v} z_i\right) \tag{7}$$

Here $N_{\mathcal{P}_n}^v$ is the number of points in voxel $v$ of the region $\mathcal{P}_n$ ($\forall n \in \{1...N\}$). Figure 3 shows the centroid approximation of the point-cloud, while Fig. 4 shows the sample voxel grids for the different regions.

### C. The Foveated Rendering Framework

Based on the system overview, the proposed *Foveated Rendering* (FR) framework, seen in Figure 5, comprises a server-client architecture that encapsulates the foveation methodology detailed in sec. II. It is divided into three parts: the **user site**, the **remote site**, and a **communication network** between them. Figure 5 shows the details and the main system components are described below:

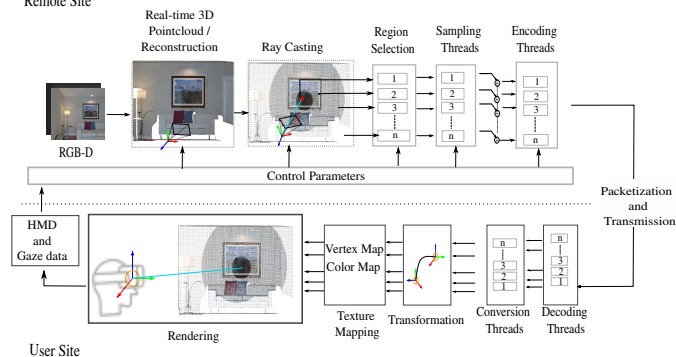

Fig. 5. Schema of the proposed *Foveated Rendering* framework.

The user site manages the: (1) decoding, conversion, and texture rendering of the streamed 3D data, (2) tracking of the eye-gaze and head-mounted display (HMD) pose, and (3) real-time transfer of gaze and pose information to the remote site. A VR-based interface is designed using the Unreal

Engine (UE) graphics development environment on Windows 10, which serves as the *IRT* environment for the user. As shown in Fig. 5 a parallel streamer, a point-cloud decoder and a conversion system to transfer the textures to the UE GPU shaders is implemented. The remote site consists of modules for acquisition, reconstruction, map partitioning, foveated sampling, encoding, and streaming, as shown in Fig. 5. A custom point-cloud and data packetization and streaming pipeline was implemented using the Boost ASIO cross-platform C++ library for the communication network.

## III. EXPERIMENT DESIGN AND EVALUATION METRICS

The experiment design focuses on an initial evaluation of the FR framework using two datasets: (i) an online synthetic dataset of a static living room environment [4], (**LIV**), seen in Fig. 4, and (ii) a dynamic scene dataset acquired using an RGB-D camera and a moving balloon (**BAL**), seen in Fig. 1.

Three test conditions were created with combinations of regions from the Table I as follows:

- **F1:** The 3D data has four partitions - Fovea, Parafovea, Perifovea, and the rest. The progressive foveated sampling in the regions follows Eq. (6). For the $4^{th}$ region, i.e., the rest of the point-cloud is sampled using the voxel sizes for the *Far Peripheral* region.
- **F2:** has five partitions - Fovea, Parafovea, and Perifovea, *Near Peripheral*, and then the rest, with a similar sampling strategy as F1.
- **F3:** includes all six partitions - Fovea, Parafovea, Perifovea, Near-, Mid-, and the Far Peripheral regions.

These conditions are compared against the reference condition **FREF**, where the FR framework is not applied on the 3D data. The following metrics were used to evaluate the FR framework: (i) *Data transfer rate* measured using the network data packet analysis tool, Wireshark [14]; (ii) End-to-end *latency* measured for each of the sub-components seen in Fig. 5; and (iii) A preliminary user study to assess the *visual quality experience* of the FR framework. Using the Double Stimulus Impairment Scale (DSIS) study approach [6] with the **LIV** dataset, subjects were first presented with the FREF condition, followed by a 3-second pause, and one of the altered conditions (F1-F3) following immediately after, in a randomized manner. The subjects were then asked to rate the second presented stimulus on a 5-point scale [6], on whether the alteration was: (5) imperceptible; (4) perceptible, but not annoying; (3) slightly annoying; (2) annoying; and (1) very annoying. 24 subjects (9 females and 15 males) participated in the study. The arithmetic mean opinion score (MOS) was calculated for each condition.

## IV. RESULTS AND CONCLUSIONS

Five randomized HMD positions with varying distances to the center of the datasets were used for the objective metrics evaluation [1]. Four hundred frames were tested for each HMD position from each dataset. Table II reports the average bandwidth and overall latency values for streaming the datasets in each condition and the relative percentage reductions in

the values as compared to the FREF condition. F1 gives an average 61% reduction against FREF. The numbers are similar for F2, while F3 offers a lower, 56% reduction. Statistical t-test analysis showed that these reductions are significant at 95% CI (p-values $<< 0.05$), against FREF. However, within the 3 conditions, the differences are not statistically significant (p-value = 0.3).

TABLE II
COMPRESSED BANDWIDTH (MBYTES/SEC; TOP ROW) AND LATENCY (MS; BOTTOM ROW) FOR REAL-TIME POINT-CLOUD STREAMING

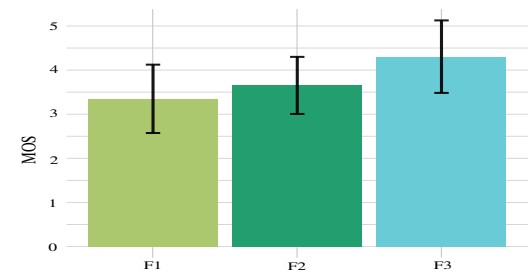

| | LIV | BAL | Reduction(%) |
|---|---|---|---|
| **F1** | 0.50 | 0.80 | |
| | 223 | 226 | |
| **F2** | 0.55 | 0.97 | |
| | 242 | 235 | |
| **F3** | 0.74 | 1.03 | |
| | 257 | 256 | |
| **FREF** | 1.32 | 1.82 | |
| | 618 | 562 | |

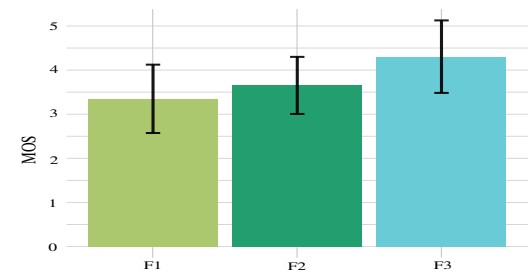

Fig. 6. Mean MOS for the VQE metric against the experimental conditions (**F1**,**F2**,**F3**). Error bars show the standard deviation.

Likewise for the latency numbers, the foveation conditions offer between 60% (F3) and 67% (F1) speedup over the FREF condition, which are statistically significant, p-values $<< 0.05$.

Figure 6 shows the MOS, averaged over the 24 subjects. It is seen that all three foveation conditions have their MOS $> 3$. For the F1 and F2 conditions, the foveation is certainly perceptible, but it may not hinder the users' experience, since the perceived degradation is only 'slightly annoying' (F1) or 'not annoying'. With an MOS $> 4$, the F3 condition shows that subjects are not able to easily perceive the degradation, and even if they do, it is 'not annoying'.

The novel FR framework presented here shows that by integrating eye-tracking, remotely acquired real-time 3D data can be represented to the user in a *foveated* way inside VR in IRT applications, which not only helps to reduce the bandwidth and latency but also does not significantly impact the visual quality experience. Future investigations will include the analysis of the limitations in the approach, e.g., effects of discontinuities at region boundaries and the over-sampling in the peripheral regions. A comprehensive user study will help situate the FR framework in terms of usability and user experience in real-world environments.

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
