# OpenReview forum: "Towards Foveated Rendering For Immersive Remote Telerobotics"
_humanrobotinteraction.org/HRI/2022/Workshop/VAM-HRI — VAM-HRI 2022_

### Official Review · Reviewer_tX39 · 2022-02-25
**Very interesting and relevant method, accept**

**Rating:** 9
**Confidence:** 5

**Review:**

The proposed method for foveated rendering in VR telerobotics is very interesting, well-written and clear, and may have a large impact on improving network latency for VR teleoperation which is very important. Overall this paper is exciting and a clear accept, although it could be improved by conducting an HRI study to evaluate the effectiveness of this method for teleoperation, as described below in the feedback:

Feedback:
1. In figure 1a, “cones” is spelt incorrectly as “cons”
2. In Section 2a, last sentence should be “this formulation applies to the….”
3. How is the color, normal, and weight information about the surfels (discussed in Section IIb) integrated into the foveated pointcloud pipeline? The same way as the surfel's positions are incorporated like in equation (3) and (7)?
4. Some of the variables mentioned in the algorithm (like L, d, dvi) should be labeled in Figure 2 to make it easier to read.
5. In order to make any claims about how useful this method is for telepresence, a user study should be conducted where the operators perform an actual task (such as identifying things in the scene, or doing a teleoperation task). It would be interesting to evaluate metrics such as how the foveated pointcloud impacts task completion time and task success.

---

### Official Review · Reviewer_Cssq · 2022-03-01
**Novel technical concept for VR visualization, accept**

**Rating:** 8
**Confidence:** 5

**Review:**

This paper presents a technique for visually displaying VR that reduces latency without significantly impacting quality of experience. "Foveation" is the concept of providing higher "resolution" closer to the area of focus and allowing regions that are out of focus to have lower clarity. The system utilizes eye gaze to determine where to apply the highest VR resolution as a way of decreasing latency. This is a fascinating concept that could have impacts for VAMHRI. Some questions follow that will hopefully help guide the workshop discussions.

- What kinds of testing will you perform to assess situational awareness, usability, and other metrics/characteristics to ensure that the reduced clarity does not affect performance?
- Are there particular HRI applications for which you anticipate this technique being especially useful?
- How would you design an HRI study to evaluate this concept for "telerobotics" as stated in your introduction?

---

### Decision · Program_Chairs · 2022-03-04

Accept